# Lookahead Optimizer: $k$ steps forward, 1 step back

**Michael R. Zhang, James Lucas, Geoffrey Hinton, Jimmy Ba**
Department of Computer Science, University of Toronto, Vector Institute
{michael, jlucas, hinton,jba}@cs.toronto.edu

## Abstract

The vast majority of successful deep neural networks are trained using variants of stochastic gradient descent (SGD) algorithms. Recent attempts to improve SGD can be broadly categorized into two approaches: (1) adaptive learning rate schemes, such as AdaGrad and Adam, and (2) accelerated schemes, such as heavy-ball and Nesterov momentum. In this paper, we propose a new optimization algorithm, Lookahead, that is orthogonal to these previous approaches and iteratively updates two sets of weights. Intuitively, the algorithm chooses a search direction by *looking ahead* at the sequence of "fast weights" generated by another optimizer. We show that Lookahead improves the learning stability and lowers the variance of its inner optimizer with negligible computation and memory cost. We empirically demonstrate Lookahead can significantly improve the performance of SGD and Adam, even with their default hyperparameter settings on ImageNet, CIFAR-10/100, neural machine translation, and Penn Treebank.

## 1   Introduction

Despite their simplicity, SGD-like algorithms remain competitive for neural network training against advanced second-order optimization methods. Large-scale distributed optimization algorithms [10, 45] have shown impressive performance in combination with improved learning rate scheduling schemes [42, 35], yet variants of SGD remain the core algorithm in the distributed systems. The recent improvements to SGD can be broadly categorized into two approaches: (1) adaptive learning rate schemes, such as AdaGrad [7] and Adam [18], and (2) accelerated schemes, such as Polyak heavy-ball [33] and Nesterov momentum [29]. Both approaches make use of the accumulated past gradient information to achieve faster convergence. However, to obtain their improved performance in neural networks often requires costly hyperparameter tuning [28].

In this work, we present Lookahead, a new optimization method, that is orthogonal to these previous approaches. Lookahead first updates the "fast weights" [12] $k$ times using any standard optimizer in its inner loop before updating the "slow weights" once in the direction of the final fast weights. We show that this update reduces the variance. We find that Lookahead is less sensitive to suboptimal hyperparameters and therefore lessens the need for extensive hyperparameter tuning. By using Lookahead with inner optimizers such as SGD or Adam, we achieve faster convergence across different deep learning tasks with minimal computational overhead.

Empirically, we evaluate Lookahead by training classifiers on the CIFAR [19] and ImageNet datasets [5], observing faster convergence on the ResNet-50 and ResNet-152 architectures [11]. We also trained LSTM language models on the Penn Treebank dataset [24] and Transformer-based [42] neural machine translation models on the WMT 2014 English-to-German dataset. For all tasks, using Lookahead leads to improved convergence over the inner optimizer and often improved generalization performance while being robust to hyperparameter changes. Our experiments demonstrate that Lookahead is robust to changes in the inner loop optimizer, the number of fast weight updates, and the slow weights learning rate.

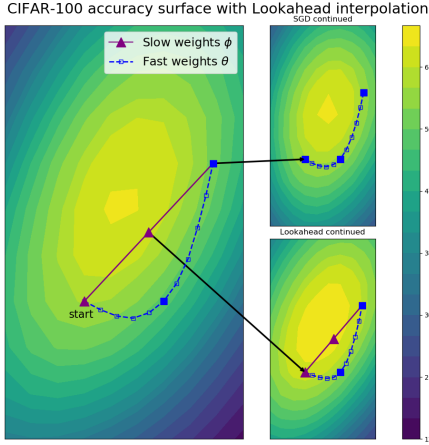

CIFAR-100 accuracy surface with Lookahead interpolation

- Slow weights $\phi$
- Fast weights $\theta$

SGD continued

Lookahead continued

start

**Algorithm 1** Lookahead Optimizer:

**Require:** Initial parameters $\phi_0$, objective function $L$
**Require:** Synchronization period $k$, slow weights step
size $\alpha$, optimizer $A$
&emsp;**for** $t = 1, 2, \ldots$ **do**
&emsp;&emsp;Synchronize parameters $\theta_{t,0} \leftarrow \phi_{t-1}$
&emsp;&emsp;**for** $i = 1, 2, \ldots, k$ **do**
&emsp;&emsp;&emsp;sample minibatch of data $d \sim \mathcal{D}$
&emsp;&emsp;&emsp;$\theta_{t,i} \leftarrow \theta_{t,i-1} + A(L, \theta_{t,i-1}, d)$
&emsp;&emsp;**end for**
&emsp;&emsp;Perform outer update $\phi_t \leftarrow \phi_{t-1} + \alpha(\theta_{t,k} - \phi_{t-1})$
&emsp;**end for**
&emsp;**return** parameters $\phi$

Figure 1: (Left) Visualizing Lookahead ($k = 10$) through a ResNet-32 test accuracy surface at epoch 100 on CIFAR-100. We project the weights onto a plane defined by the first, middle, and last fast (inner-loop) weights. The fast weights are along the blue dashed path. All points that lie on the plane are represented as solid, including the entire Lookahead slow weights path (in purple). Lookahead (middle, bottom right) quickly progresses closer to the minima than SGD (middle, top right) is able to. (Right) Pseudocode for Lookahead.

## 2  Method

In this section, we describe the Lookahead algorithm and discuss its properties. Lookahead maintains a set of slow weights $\phi$ and fast weights $\theta$, which get synced with the fast weights every $k$ updates. The fast weights are updated through applying $A$, any standard optimization algorithm, to batches of training examples sampled from the dataset $\mathcal{D}$. After $k$ inner optimizer updates using $A$, the slow weights are updated towards the fast weights by linearly interpolating in weight space, $\theta - \phi$. We denote the slow weights learning rate as $\alpha$. After each slow weights update, the fast weights are reset to the current slow weights value. Psuedocode is provided in Algorithm 1.[1]

Standard optimization methods typically require carefully tuned learning rates to prevent oscillation and slow convergence. This is even more important in the stochastic setting [25, 43]. Lookahead, however, benefits from a larger learning rate in the inner loop. When oscillating in the high curvature directions, the fast weights updates make rapid progress along the low curvature directions. The slow weights help smooth out the oscillations through the parameter interpolation. The combination of fast weights and slow weights improves learning in high curvature directions, reduces variance, and enables Lookahead to converge rapidly in practice.

Figure 1 shows the trajectory of both the fast weights and slow weights during the optimization of a ResNet-32 model on CIFAR-100. While the fast weights explore around the minima, the slow weight update pushes Lookahead aggressively towards an area of improved test accuracy, a region which remains unexplored by SGD after 20 updates.

**Slow weights trajectory** We can characterize the trajectory of the slow weights as an exponential moving average (EMA) of the *final* fast weights within each inner-loop, regardless of the inner optimizer. After $k$ inner-loop steps we have:

$$\phi_{t+1} = \phi_t + \alpha(\theta_{t,k} - \phi_t) \tag{1}$$

$$= \alpha[\theta_{t,k} + (1-\alpha)\theta_{t-1,k} + \ldots + (1-\alpha)^{t-1}\theta_{0,k}] + (1-\alpha)^t\phi_0 \tag{2}$$

Intuitively, the slow weights heavily utilize recent proposals from the fast weight optimization but maintain some influence from previous fast weights. We show that this has the effect of reducing variance in Section 3.1. While a Polyak-style average has further theoretical guarantees, our results match the claim that "an exponentially-decayed moving average typically works much better in practice" [25].

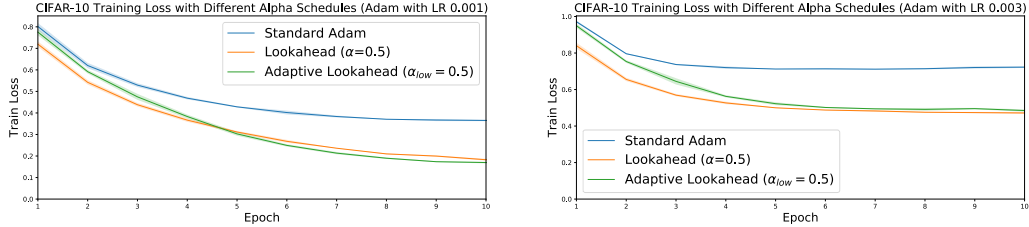

Figure 2: CIFAR-10 training loss with fixed and adaptive $\alpha$. The adaptive $\alpha$ is clipped between $[\alpha_{low}, 1]$. (Left) Adam learning rate = 0.001. (Right) Adam learning rate = 0.003.

**Fast weights trajectory**  Within each inner-loop, the trajectory of the fast weights depends on the choice of underlying optimizer. Given an optimization algorithm $A$ that takes in an objective function $L$ and the current mini-batch training examples $d$, we have the update rule for the fast weights:

$$\theta_{t,i+1} = \theta_{t,i} + A(L, \theta_{t,i-1}, d). \tag{3}$$

We have the choice of maintaining, interpolating, or resetting the internal state (e.g. momentum) of the inner optimizer. We evaluate this tradeoff on the CIFAR dataset (where every choice improves convergence) in Appendix D.1 and maintain internal state for the other experiments.

**Computational complexity**  Lookahead has a constant computational overhead due to parameter copying and basic arithmetic operations that is amortized across the $k$ inner loop updates. The number of operations is $\mathcal{O}(\frac{k+1}{k})$ times that of the inner optimizer. Lookahead maintains a single additional copy of the number of learnable parameters in the model.

## 2.1  Selecting the Slow Weights Step Size

The step size in the direction $(\theta_{t,k} - \theta_{t,0})$ is controlled by $\alpha$. By taking a quadratic approximation of the loss, we present a principled way of selecting $\alpha$.

**Proposition 1** (Optimal slow weights step size). *For a quadratic loss function $L(x) = \frac{1}{2}x^T A x - b^T x$, the step size $\alpha^*$ that minimizes the loss for two points $\theta_{t,0}$ and $\theta_{t,k}$ is given by:*

$$\alpha^* = \arg\min_{\alpha} L(\theta_{t,0} + \alpha(\theta_{t,k} - \theta_{t,0})) = \frac{(\theta_{t,0} - \theta^*)^T A(\theta_{t,0} - \theta_{t,k})}{(\theta_{t,0} - \theta_{t,k})^T A(\theta_{t,0} - \theta_{t,k})}$$

*where $\theta^* = A^{-1}b$ minimizes the loss.*

Proof is in the appendix. Using quadratic approximations for the curvature, which is typical in second order optimization [7, 18, 26], we can derive an estimate for the optimal $\alpha$ more generally. The full Hessian is typically intractable so we instead use aforementioned approximations, such as the diagonal approximation to the empirical Fisher used by the Adam optimizer [18]. This approximation works well in our numerical experiments if we clip the magnitude of the step size. At each slow weight update, we compute:

$$\hat{\alpha}^* = \text{clip}(\frac{(\theta_{t,0} - (\theta_{t,k} - \hat{A}^{-1}\hat{\nabla}L(\theta_{t,k}))^T \hat{A}(\theta_{t,0} - \theta_{t,k})}{(\theta_{t,0} - \theta_{t,k})^T \hat{A}(\theta_{t,0} - \theta_{t,k})}, \alpha_{\text{low}}, 1)$$

where $\hat{A}$ is the empirical Fisher approximation and $\theta_{t,k} - \hat{A}^{-1}\hat{\nabla}L(\theta_{t,k})$ approximates the optimum $\theta^*$. We prove Proposition 1 and elaborate on assumptions in the appendix B.2. Setting $\alpha_{\text{low}} > 0$ improves the stability of our algorithm. We evaluate the performance of this adaptive scheme versus a fixed scheme and standard Adam on a ResNet-18 trained on CIFAR-10 with two different learning rates and show the results in Figure 2. Additional hyperparameter details are given in appendix C. Both the fixed and adaptive Lookahead offer improved convergence.

In practice, a fixed choice of $\alpha$ offers similar convergence benefits and tends to generalize better. Fixing $\alpha$ avoids the need to maintain an estimate of the empirical Fisher, which incurs a memory and computational cost when the inner optimizer does not maintain such an estimate e.g. SGD. We thus use a fixed $\alpha$ for the rest of our deep learning experiments.

# 3 Convergence Analysis

## 3.1 Noisy quadratic analysis

We analyze Lookahead on a noisy quadratic model to better understand its convergence guarantees. While simple, this model is a proxy for neural network optimization and effectively optimizing it remains a challenging open problem [37, 26, 43, 47]. In this section, we will show under equal learning rates that Lookahead will converge to a smaller steady-state risk than SGD. We will then show through simulation of the expected dynamics that Lookahead is able to converge to this steady-state risk more quickly than SGD for a range of hyperparameter settings.

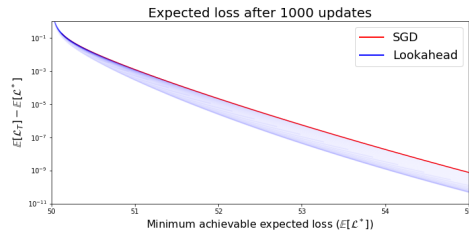

Figure 3: Comparing expected optimization progress between SGD and Lookahead($k = 5$) on the noisy quadratic model. Each vertical slice compares the convergence of optimizers with the same final loss values. For Lookahead, convergence rates for 100 evenly spaced $\alpha$ values in the range $(0, 1]$ are overlaid.

**Model definition**   We use the same model as in Schaul et al. [37] and Wu et al. [43].

$$\hat{\mathcal{L}}(\mathbf{x}) = \frac{1}{2}(\mathbf{x} - \mathbf{c})^T \mathbf{A}(\mathbf{x} - \mathbf{c}), \qquad (4)$$

with $\mathbf{c} \sim \mathcal{N}(\mathbf{x}^*, \Sigma)$. We assume that both $\mathbf{A}$ and $\Sigma$ are diagonal and that, without loss of generality, $\mathbf{x}^* = \mathbf{0}$. While it is trivial to assume that $\mathbf{A}$ is diagonal [2] the co-diagonalizable noise assumption is non-trivial but is common — see Wu et al. [43] and Zhang et al. [47] for further discussion. We use $a_i$ and $\sigma_i^2$ to denote the diagonal elements of $\mathbf{A}$ and $\Sigma$ respectively. Taking the expectation over $\mathbf{c}$, the expected loss of the iterates $\theta^{(t)}$ is,

$$\mathcal{L}(\theta^{(t)}) = \mathbb{E}[\hat{\mathcal{L}}(\theta^{(t)})] = \frac{1}{2}\mathbb{E}[\sum_i a_i(\theta_i^{(t)^2} + \sigma_i^2)] = \frac{1}{2}\sum_i a_i(\mathbb{E}[\theta_i^{(t)}]^2 + \mathbb{V}[\theta_i^{(t)}] + \sigma_i^2). \quad (5)$$

Analyzing the expected dynamics of the SGD iterates and the slow weights gives the following result.

**Proposition 2** (Lookahead steady-state risk). *Let $0 < \gamma < 2/L$ be the learning rate of SGD and Lookahead where $L = \max_i a_i$. In the noisy quadratic model, the iterates of SGD and Lookahead with SGD as its inner optimizer converge to $0$ in expectation and the variances converge to the following fixed points:*

$$V_{SGD}^* = \frac{\gamma^2 \mathbf{A}^2 \Sigma^2}{\mathbf{I} - (\mathbf{I} - \gamma \mathbf{A})^2} \qquad (6)$$

$$V_{LA}^* = \frac{\alpha^2(\mathbf{I} - (\mathbf{I} - \gamma \mathbf{A})^{2k})}{\alpha^2(\mathbf{I} - (\mathbf{I} - \gamma \mathbf{A})^{2k}) + 2\alpha(1-\alpha)(\mathbf{I} - (\mathbf{I} - \gamma \mathbf{A})^k)} V_{SGD}^* \qquad (7)$$

**Remarks**   For the Lookahead variance fixed point, the first product term is always smaller than 1 for $\alpha \in (0, 1)$, and thus Lookahead has a variance fixed point that is strictly smaller than that of the SGD inner-loop optimizer for the same learning rate. Evidence of this phenomenon is present in deep neural networks trained on the CIFAR dataset, shown in Figure 10.

In Proposition 2, we use the same learning rate for both SGD and Lookahead. To fairly evaluate the convergence of the two methods, we compare the convergence rates under hyperparameter settings that achieve the same steady-state risk. In Figure 3 we show the expected loss after 1000 updates (computed analytically) for both Lookahead and SGD. This shows that there exists (fixed) settings of the Lookahead hyperparameters that arrive at the same steady state risk as SGD but do so more quickly. Moreover, Lookahead outperforms SGD across the broad spectrum of $\alpha$ values we simulated. Details, further simulation results, and additional discussion are presented in Appendix B.

## 3.2 Deterministic quadratic convergence

In the previous section we showed that on the noisy quadratic model, Lookahead is able to improve convergence of the SGD optimizer under setting with equivalent convergent risk. Here we analyze the quadratic model without noise using gradient descent with momentum [33, 9] and show that when the system is under-damped, Lookahead is able to improve on the convergence rate.

As before, we restrict our attention to diagonal quadratic functions (which in this case is entirely without loss of generality). Given an initial point $\boldsymbol{\theta}_0$, we wish to find the rate of contraction, that is, the smallest $\rho$ satisfying $||\boldsymbol{\theta}_t - \boldsymbol{\theta}^*|| \leq \rho^t ||\boldsymbol{\theta}_0 - \boldsymbol{\theta}^*||$. We follow the approach of [31] and model the optimization of this function as a linear dynamical system allowing us to compute the rate exactly. Details are in Appendix B.

As in Lucas et al. [23], to better understand the sensitivity of Lookahead to misspecified conditioning we fix the momentum coefficient of classical momentum and explore the convergence rate over varying condition number under the optimal learning rate. As expected, Lookahead has slightly worse convergence in the over-damped regime where momentum is set too low and CM is slowly, monotonically converging to the optimum. However, when the system is under-damped (and oscillations occur) Lookahead is able to significantly improve the convergence rate by skipping to a better parameter setting during oscillation.

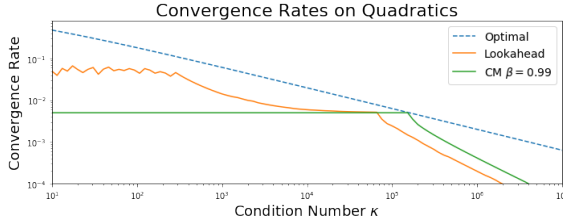

Figure 4: Quadratic convergence rates $(1 - \rho)$ of classical momentum versus Lookahead wrapping classical momentum. For Lookahead, we fix $k = 20$ lookahead steps and $\alpha = 0.5$ for the slow weights step size. Lookahead is able to significantly improve on the convergence rate in the under-damped regime where oscillations are observed.

## 4 Related work

Our work is inspired by recent advances in understanding the loss surface of deep neural networks. While the idea of following the trajectory of weights dates back to Ruppert [36], Polyak and Juditsky [34], averaging weights in neural networks has not been carefully studied until more recently. Garipov et al. [8] observe that the final weights of two independently trained neural networks can be connected by a curve with low loss. Izmailov et al. [14] proposes Stochastic Weight Averaging (SWA), which averages the weights at different checkpoints obtained during training. Parameter averaging schemes are used to create ensembles in natural language processing tasks [15, 27] and in training Generative Adversarial Networks [44]. In contrast to previous approaches, which generally focus on generating a set of parameters at the *end* of training, Lookahead is an optimization algorithm which performs parameter averaging *during* the training procedure to achieve faster convergence. We elaborate on differences with SWA and present additional experimental results in appendix D.3.

The Reptile algorithm, proposed by Nichol et al. [30], samples tasks in its outer loop and runs an optimization algorithm on each task within the inner loop. The initial weights are then updated in the direction of the new weights. While the functionality is similar, the application and setting are starkly different. Reptile samples different tasks and aims to find parameters which act as good initial values for new tasks sampled at test time. Lookahead does not sample new tasks for each outer loop and aims to take advantage of the geometry of loss surfaces to improve convergence.

Katyusha [1], an accelerated form of SVRG [17], also uses an outer and inner loop during optimization. Katyusha checkpoints parameters during optimization. Within each inner loop step, the parameters are pulled back towards the latest checkpoint. Lookahead computes the pullback only at the end of the inner loop and the gradient updates do not utilize the SVRG correction (though this would be possible). While Katyusha has theoretical guarantees in the convex optimization setting, the SVRG-based update does not work well for neural networks [4].

Anderson acceleration [2] and other related extrapolation techniques [3] have a similar flavor to Lookahead. These methods keep track of all iterates within an inner loop and then compute some linear combination which extrapolates the iterates towards their fixed point. This presents additional

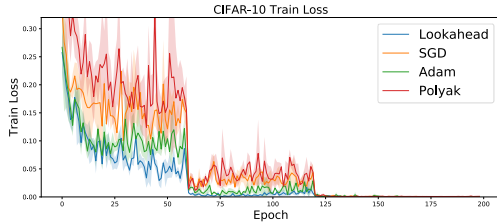

| OPTIMIZER | CIFAR-10 | CIFAR-100 |
|---|---|---|
| SGD | $95.23 \pm .19$ | $78.24 \pm .18$ |
| POLYAK | $95.26 \pm .04$ | $77.99 \pm .42$ |
| ADAM | $94.84 \pm .16$ | $76.88 \pm .39$ |
| LOOKAHEAD | $95.27 \pm .06$ | $78.34 \pm .05$ |

Table 1: CIFAR Final Validation Accuracy.

Figure 5: Performance comparison of the different optimization algorithms. (**Left**) Train Loss on CIFAR-100. (**Right**) CIFAR ResNet-18 validation accuracies with various optimizers. We do a grid search over learning rate and weight decay on the other optimizers (details in appendix C). Lookahead and Polyak are wrapped around SGD.

challenges first in the form of additional memory overhead as the number of inner-loop steps increases and also in finding the best linear combination. Scieur et al. [38, 39] propose a method by which to find a good linear combination and apply this approach to deep learning problems and report both improved convergence and generalization. However, their method requires on the order of $k$ times more memory than Lookahead. Lookahead can be seen as a simple version of Anderson acceleration wherein only the first and last iterates are used.

## 5 Experiments

We completed a thorough evaluation of the Lookahead optimizer on a variety of deep learning tasks against well-calibrated baselines. We explored image classification on CIFAR-10/CIFAR-100 [19] and ImageNet [5]. We also trained LSTM language models on the Penn Treebank dataset [24] and Transformer-based [42] neural machine translation models on the WMT 2014 English-to-German dataset. For all of our experiments, every algorithm consumed the same amount of training data.

### 5.1 CIFAR-10 and CIFAR-100

The CIFAR-10 and CIFAR-100 datasets for classification consist of $32 \times 32$ color images, with 10 and 100 different classes, split into a training set with 50,000 images and a test set with 10,000 images. We ran all our CIFAR experiments with 3 seeds and trained for 200 epochs on a ResNet-18 [11] with batches of 128 images and decay the learning rate by a factor of 5 at the 60th, 120th, and 160th epochs. Additional details are given in appendix C.

We summarize our results in Figure 5.[3] We also elaborate on how Lookahead contrasts with SWA and present results demonstrating lower validation error with Pre-ResNet-110 and Wide-ResNet-28-10 [46] on CIFAR-100 in appendix D.3. Note that Lookahead achieves significantly faster convergence throughout training even though the learning rate schedule is optimized for the inner optimizer—future work can involve building a learning rate schedule for Lookahead. This improved convergence is important for better anytime performance in new datasets where hyperparameters and learning rate schedules are not well-calibrated.

### 5.2 ImageNet

The 1000-way ImageNet task [5] is a classification task that contains roughly 1.28 million training images and 50,000 validation images. We use the official PyTorch implementation[4] and the ResNet-50 and ResNet-152 [11] architectures. Our baseline algorithm is SGD with an initial learning rate of 0.1 and momentum value of 0.9. We train for 90 epochs and decay our learning rate by a factor of 10 at the 30th and 60th epochs. For Lookahead, we set $k = 5$ and slow weights step size $\alpha = 0.5$.

Motivated by the improved convergence we observed in our initial experiment, we tried a more aggressive learning rate decay schedule where we decay the learning rate by a factor of 10 at the 30th, 48th, and 58th epochs. Using such a schedule, we reach $75\%$ single crop top-1 accuracy on ImageNet in just 50 epochs and reach $75.5\%$ top-1 accuracy in 60 epochs. The results are shown in Figure 6.

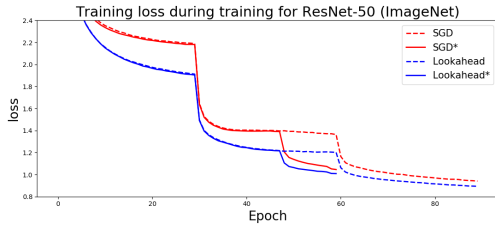

| OPTIMIZER | LA | SGD |
|---|---|---|
| EPOCH 50 - TOP 1 | 75.13 | 74.43 |
| EPOCH 50 - TOP 5 | 92.22 | 92.15 |
| EPOCH 60 - TOP 1 | 75.49 | 75.15 |
| EPOCH 60 - TOP 5 | 92.53 | 92.56 |

Table 2: Top-1 and Top-5 single crop validation accuracies on ImageNet.

Figure 6: ImageNet training loss. The asterisk denotes the aggressive learning rate decay schedule, where LR is decayed at iteration 30, 48, and 58. We report validation accuracies for this schedule.

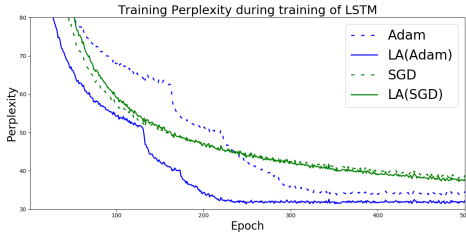

(a) Training perplexity of LSTM models trained on the Penn Treebank dataset

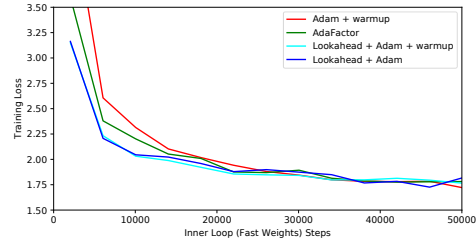

(b) Training Loss on Transformer. Adam and AdaFactor both use a linear warmup scheme described in Vaswani et al. [42].

Figure 7: Optimization performance on Penn Treebank and WMT-14 machine translation task.

To test the scalability of our method, we ran Lookahead with the aggressive learning rate decay on ResNet-152. We reach 77% single crop top-1 accuracy in 49 epochs (matching what is reported in He et al. [11]) and 77.96% top-1 accuracy in 60 epochs. Other approaches for improving convergence on ImageNet can require hundreds of GPUs, or tricks such as ramping up the learning rate and adaptive batch-sizes [10, 16]. The fastest convergence we are aware of uses an approximate second-order method to train a ResNet-50 to 75% top-1 accuracy in 35 epochs with 1,024 GPUs [32]. In contrast, Lookahead requires changing one single line of code and can easily scale to ResNet-152.

### 5.3 Language modeling

We trained LSTMs [13] for language modeling on the Penn Treebank dataset. We followed the model setup of Merity et al. [27] and made use of their publicly available code in our experiments. We did not include the fine-tuning stages. We searched over hyperparameters for both Adam and SGD (without momentum) to find the model which gave the best validation performance. We then performed an additional small grid search on each of these methods with Lookahead. Each model was trained for 750 epochs. We show training curves for each model in Figure 7a.

Using Lookahead with Adam we were able to achieve the fastest convergence and best training, validation, and test perplexity. The models trained with SGD took much longer to converge (around 700 epochs) and were unable to match the final performance of Adam. Using Polyak weight averaging [34] with SGD, as suggested by Merity et al. [27] and referred to as ASGD, we were able to improve on the performance of Adam but were unable to match the performance of Lookahead. Full results are given in Table 3 and additional details are in appendix C.

### 5.4 Neural machine translation

We trained Transformer based models [42] on the WMT2014 English-to-German translation task on a single Tensor Processing Unit (TPU) node. We took the base model from Vaswani et al. [42] and trained it using the proposed warmup-then-decay learning rate scheduling scheme and, additionally, the same scheme wrapped with Lookahead. We found Lookahead speedups the early stage of the training over Adam and the later proposed AdaFactor [40] optimizer. All the methods converge to similar training loss and BLEU score at the end, see Figure 7b and Table 4.

Table 3: LSTM training, validation, and test perplexity on the Penn Treebank dataset.

| OPTIMIZER | TRAIN | VAL. | TEST |
|---|---|---|---|
| SGD | 43.62 | 66.0 | 63.90 |
| LA(SGD) | 35.02 | 65.10 | 63.04 |
| ADAM | 33.54 | 61.64 | 59.33 |
| LA(ADAM) | **31.92** | **60.28** | **57.72** |
| POLYAK | - | 61.18 | 58.79 |

Table 4: Transformer Base Model trained for 50k steps on WMT English-to-German. "Adam-" denote Adam without learning rate warm-up.

| OPTIMIZER | NEWSTEST13 | NEWSTEST14 |
|---|---|---|
| ADAM | 24.6 | 24.6 |
| LA(ADAM) | 24.68 | 24.70 |
| LA(ADAM-) | 24.3 | 24.4 |
| ADAFACTOR | 24.17 | 24.51 |

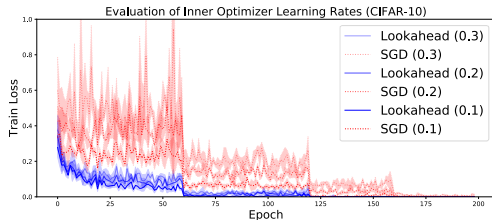

(a) CIFAR-10 Train Loss: Different LR

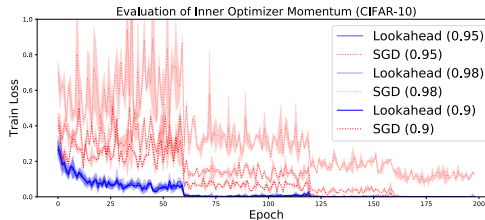

(b) CIFAR-10 Train Loss: Different momentum

Figure 8: We fix Lookahead parameters and evaluate on different inner optimizers.

Our NMT experiments further confirms Lookahead improves the robustness of the inner loop optimizer. We found Lookahead enables a wider range of learning rate {0.02, 0.04, 0.06} choices for the Transformer model that all converge to similar final losses. Full details are given in Appendix C.4.

### 5.5 Empirical analysis

**Robustness to inner optimization algorithm, $k$, and $\alpha$** We demonstrate empirically on the CIFAR dataset that Lookahead consistently delivers fast convergence across different hyperparameter settings. We fix slow weights step size $\alpha = 0.5$ and $k = 5$ and run Lookahead on inner SGD optimizers with different learning rates and momentum; results are shown in Figure 8. In general, we observe that Lookahead can train with higher learning rates on the base optimizer with little to no tuning on $k$ and $\alpha$. This agrees with our discussion of variance reduction in Section 3.1. We also evaluate robustness to the Lookahead hyperparameters by fixing the inner optimizer and evaluating runs with varying updates $k$ and step size $\alpha$; these results are shown in Figure 9.

**Inner loop and outer loop evaluation** To get a better understanding of the Lookahead update, we also plotted the test accuracy for every update on epoch 65 in Figure 10. We found that within each inner loop the fast weights may lead to substantial degradation in task performance—this reflects our analysis of the higher variance of the inner loop update in section 3.1. The slow weights step recovers the outer loop variance and restores the test accuracy.

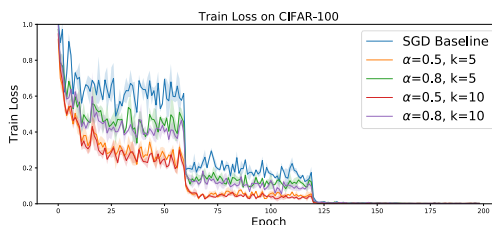

| $\alpha$ $\kappa$ | 0.5 | 0.8 |
|---|---|---|
| 5 | 78.24 ± .02 | 78.27 ± .04 |
| 10 | 78.19 ± .22 | 77.94 ± .22 |

Table 5: All settings have higher validation accuracy than SGD (77.72%)

Figure 9: CIFAR-100 train loss and final test accuracy with various $k$ and $\alpha$.

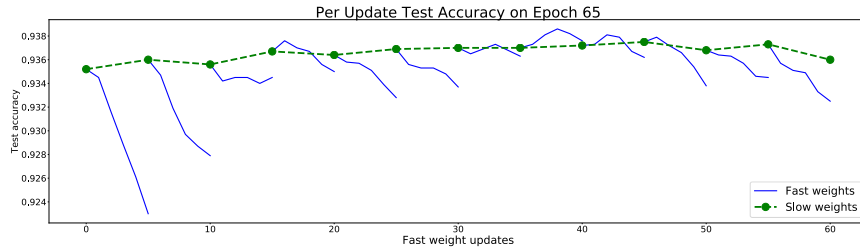

Figure 10: Visualizing Lookahead accuracy for 60 fast weight updates. We plot the test accuracy after every update (the training accuracy and loss behave similarly). The inner loop update tends to degrade both the training and test accuracy, while the interpolation recovers the original performance.

## 6  Conclusion

In this paper, we present Lookahead, an algorithm that can be combined with any standard optimization method. Our algorithm computes weight updates by *looking ahead* at the sequence of "fast weights" generated by another optimizer. We illustrate how Lookahead improves convergence by reducing variance and show strong empirical results on many deep learning benchmark datasets and architectures.

## Acknowledgements

We'd like to thank Roger Grosse, Guodong Zhang, Denny Wu, Silviu Pitis, David Madras, Jackson Wang, Harris Chan, and Mufan Li for helpful comments on earlier versions of this work. We are also thankful for the many helpful comments from anonymous reviewers.

## Footnotes

[1] Our open source implementation is available at https://github.com/michaelrzhang/lookahead.

[2]Classical momentum's iterates are invariant to translations and rotations (see e.g. Sutskever et al. [41]) and Lookahead's linear interpolation is also invariant to such changes.

[3] We refer to SGD with heavy ball momentum [33] as SGD.

[4] Implementation available at `https://github.com/pytorch/examples/tree/master/imagenet`.

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
