[Supplementary Material]

# Lookahead Optimizer: $k$ steps forward, 1 step back

**Michael R. Zhang, James Lucas, Geoffrey Hinton, Jimmy Ba**
Department of Computer Science, University of Toronto, Vector Institute
{michael, jlucas, hinton,jba}@cs.toronto.edu

## Abstract

The vast majority of successful deep neural networks are trained using variants of stochastic gradient descent (SGD) algorithms. Recent attempts to improve SGD can be broadly categorized into two approaches: (1) adaptive learning rate schemes, such as AdaGrad and Adam, and (2) accelerated schemes, such as heavy-ball and Nesterov momentum. In this paper, we propose a new optimization algorithm, Lookahead, that is orthogonal to these previous approaches and iteratively updates two sets of weights. Intuitively, the algorithm chooses a search direction by *looking ahead* at the sequence of "fast weights" generated by another optimizer. We show that Lookahead improves the learning stability and lowers the variance of its inner optimizer with negligible computation and memory cost. We empirically demonstrate Lookahead can significantly improve the performance of SGD and Adam, even with their default hyperparameter settings on ImageNet, CIFAR-10/100, neural machine translation, and Penn Treebank.

## 1  Introduction

Despite their simplicity, SGD-like algorithms remain competitive for neural network training against advanced second-order optimization methods. Large-scale distributed optimization algorithms [10, 45] have shown impressive performance in combination with improved learning rate scheduling schemes [42, 35], yet variants of SGD remain the core algorithm in the distributed systems. The recent improvements to SGD can be broadly categorized into two approaches: (1) adaptive learning rate schemes, such as AdaGrad [7] and Adam [18], and (2) accelerated schemes, such as Polyak heavy-ball [33] and Nesterov momentum [29]. Both approaches make use of the accumulated past gradient information to achieve faster convergence. However, to obtain their improved performance in neural networks often requires costly hyperparameter tuning [28].

In this work, we present Lookahead, a new optimization method, that is orthogonal to these previous approaches. Lookahead first updates the "fast weights" [12] $k$ times using any standard optimizer in its inner loop before updating the "slow weights" once in the direction of the final fast weights. We show that this update reduces the variance. We find that Lookahead is less sensitive to suboptimal hyperparameters and therefore lessens the need for extensive hyperparameter tuning. By using Lookahead with inner optimizers such as SGD or Adam, we achieve faster convergence across different deep learning tasks with minimal computational overhead.

Empirically, we evaluate Lookahead by training classifiers on the CIFAR [19] and ImageNet datasets [5], observing faster convergence on the ResNet-50 and ResNet-152 architectures [11]. We also trained LSTM language models on the Penn Treebank dataset [24] and Transformer-based [42] neural machine translation models on the WMT 2014 English-to-German dataset. For all tasks, using Lookahead leads to improved convergence over the inner optimizer and often improved generalization performance while being robust to hyperparameter changes. Our experiments demonstrate that Lookahead is robust to changes in the inner loop optimizer, the number of fast weight updates, and the slow weights learning rate.

CIFAR-100 accuracy surface with Lookahead interpolation

SGD continued

Slow weights $\phi$
Fast weights $\theta$

Lookahead continued

start

---

**Algorithm 1** Lookahead Optimizer:

**Require:** Initial parameters $\phi_0$, objective function $L$
**Require:** Synchronization period $k$, slow weights step
    size $\alpha$, optimizer $A$
    **for** $t = 1, 2, \ldots$ **do**
        Synchronize parameters $\theta_{t,0} \leftarrow \phi_{t-1}$
        **for** $i = 1, 2, \ldots, k$ **do**

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

# A Noisy quadratic analysis

Here we present the details of the noisy quadratic analysis, and the proof of Proposition 2.

**Stochastic dynamics of SGD** From Wu et al. [43], we can compute the dynamics of SGD with learning rate $\gamma$ as follows:

$$\mathbb{E}[\mathbf{x}^{(t+1)}] = (\mathbf{I} - \gamma\mathbf{A})\,\mathbb{E}[\mathbf{x}^{(t)}] \tag{8}$$

$$\mathbb{V}[\mathbf{x}^{(t+1)}] = (\mathbf{I} - \gamma\mathbf{A})^2\,\mathbb{V}[\mathbf{x}^{(t)}] + \gamma^2\mathbf{A}^2\Sigma \tag{9}$$

**Stochastic dynamics of Lookahead SGD** We now compute the dynamics of the slow weights of Lookahead.

**Lemma 1.** *The Lookahead slow weights have the following trajectories:*

$$\mathbb{E}[\boldsymbol{\phi}_{t+1}] = [1 - \alpha + \alpha(\mathbf{I} - \gamma\mathbf{A})^k]\,\mathbb{E}[\boldsymbol{\phi}_t] \tag{10}$$

$$\mathbb{V}[\boldsymbol{\phi}_{t+1}] = [1 - \alpha + \alpha(\mathbf{I} - \gamma\mathbf{A})^k]^2\,\mathbb{V}[\boldsymbol{\phi}_t] + \alpha^2 \sum_{i=0}^{k-1}(\mathbf{I} - \gamma\mathbf{A})^{2i}\gamma^2\mathbf{A}^2\Sigma \tag{11}$$

*Proof.* The expectation trajectory follows from SGD,

$$\begin{aligned}
\mathbb{E}[\boldsymbol{\phi}_{t+1}] &= (1 - \alpha)\,\mathbb{E}[\boldsymbol{\phi}_t] + \alpha\,\mathbb{E}[\boldsymbol{\theta}_{t,k}] \\
&= (1 - \alpha)\,\mathbb{E}[\boldsymbol{\phi}_t] + \alpha(\mathbf{I} - \gamma\mathbf{A})^k\,\mathbb{E}[\boldsymbol{\phi}_t] \\
&= [1 - \alpha + \alpha(\mathbf{I} - \gamma\mathbf{A})^k]\,\mathbb{E}[\boldsymbol{\phi}_t]
\end{aligned}$$

For the variance, we can write $\mathbb{V}[\boldsymbol{\phi}_{t+1}] = (1 - \alpha)^2\,\mathbb{V}[\boldsymbol{\phi}_t] + \alpha^2\,\mathbb{V}[\boldsymbol{\theta}_{t,k}] + 2\alpha(1 - \alpha)\mathrm{cov}(\boldsymbol{\phi}_t, \boldsymbol{\theta}_{t,k})$. We proceed by computing the covariance term recursively. For simplicity, we work with a single element, $\theta$, of the vector $\boldsymbol{\theta}$ (as $\mathbf{A}$ is diagonal, each element evolves independently).

$$\begin{aligned}
\mathrm{cov}(\theta_{t,k-1}, \theta_{t,k}) &= \mathbb{E}[(\theta_{t,k-1} - \mathbb{E}[\theta_{t,k-1}])(\theta_{t,k} - \mathbb{E}[\theta_{t,k}])] \\
&= \mathbb{E}[(\theta_{t,k-1} - \mathbb{E}[\theta_{t,k-1}])(\theta_{t,k} - (1 - \gamma a)\,\mathbb{E}[\theta_{t,k-1}])] \\
&= \mathbb{E}[\theta_{t,k-1}\theta_{t,k}] - (1 - \gamma a)\,\mathbb{E}[\theta_{t,k-1}]^2 \\
&= \mathbb{E}[(1 - \gamma a)\theta_{t,k-1}^2] - (1 - \gamma a)\,\mathbb{E}[\theta_{t,k-1}]^2 \\
&= (1 - \gamma a)\,\mathbb{V}[\theta_{t,k-1}]
\end{aligned}$$

A similar derivation yields $\mathrm{cov}(\boldsymbol{\phi}_t, \boldsymbol{\theta}_{t,k}) = (\mathbf{I} - \gamma\mathbf{A})^k\,\mathbb{V}[\boldsymbol{\phi}_t]$. After substituting the SGD variance formula and some rearranging we have,

$$\mathbb{V}[\boldsymbol{\phi}_{t+1}] = [1 - \alpha + \alpha(\mathbf{I} - \gamma\mathbf{A})^k]^2\,\mathbb{V}[\boldsymbol{\phi}_t] + \alpha^2 \sum_{i=0}^{k-1}(\mathbf{I} - \gamma\mathbf{A})^{2i}\gamma^2\mathbf{A}^2\Sigma$$

$\square$

We now proceed with the proof of Proposition 2.

*Proof.* First note that if the learning rate is chosen as specified, then each of the trajectories is a contraction map. By Banach's fixed point theorem, they each have a unique fixed point. Clearly the expectation trajectories contract to zero in each case.

For the variance we can solve for the fixed points directly. For SGD,

$$V_{SGD}^* = (1 - \gamma\mathbf{A})^2 V_{SGD}^* + \gamma\mathbf{A}^2\Sigma,$$

$$\Rightarrow V_{SGD}^* = \frac{\gamma^2\mathbf{A}^2\Sigma}{\mathbf{I} - (\mathbf{I} - \gamma\mathbf{A})^2}.$$

For Lookahead, we have,

$$V_{LA}^* = [1 - \alpha + \alpha(\mathbf{I} - \gamma\mathbf{A})^k]^2 V_{LA}^* + \alpha^2 \sum_{i=0}^{k-1} (\mathbf{I} - \gamma\mathbf{A})^{2i} \gamma^2 \mathbf{A}^2 \Sigma$$

$$\Rightarrow V_{LA}^* = \frac{\alpha^2 \sum_{i=0}^{k-1} (\mathbf{I} - \gamma\mathbf{A})^{2i}}{\mathbf{I} - [(1-\alpha)\mathbf{I} + \alpha(\mathbf{I} - \gamma\mathbf{A})^k]^2} \gamma^2 \mathbf{A}^2 \Sigma$$

$$\Rightarrow V_{LA}^* = \frac{\alpha^2 (\mathbf{I} - (\mathbf{I} - \gamma\mathbf{A})^{2k})}{\mathbf{I} - [(1-\alpha)\mathbf{I} + \alpha(\mathbf{I} - \gamma\mathbf{A})^k]^2} \frac{\gamma^2 \mathbf{A}^2 \Sigma}{\mathbf{I} - (\mathbf{I} - \gamma\mathbf{A})^2}$$

where for the final equality, we used the identity $\sum_0^k a^i = (1 - a^k)/(1 - a)$. Some standard manipulations of the denominator on the first term lead to the final solution,

$$V_{LA}^* = \frac{\alpha^2 (\mathbf{I} - (\mathbf{I} - \gamma\mathbf{A})^{2k})}{\alpha^2 (\mathbf{I} - (\mathbf{I} - \gamma\mathbf{A})^{2k}) + 2\alpha(1 - \alpha)(\mathbf{I} - (\mathbf{I} - \gamma\mathbf{A})^k)} \frac{\gamma^2 \mathbf{A}^2 \Sigma^2}{\mathbf{I} - (\mathbf{I} - \gamma\mathbf{A})^2}$$

$\square$

For the same learning rate, Lookahead will achieve a smaller loss as the variance is reduced more. However, the convergence speed of the expectation term will be slower as we must compare $1 - \alpha + \alpha(\mathbf{I} - \gamma\mathbf{A})^k$ to $(\mathbf{I} - \gamma\mathbf{A})^k$ and the latter is always smaller for $\alpha < 1$. In our experiments, we observe that Lookahead typically converges much faster than its inner optimizer. We speculate that the learning rate for the inner optimizer is set sufficiently high such that the variance reduction term is more important–this is the more common regime for neural networks that attain high validation accuracy, as higher initial learning rates are used to overcome the short-horizon bias [43].

### A.1 Comparing convergence rates

In Figure 3 we compared the convergence rates of SGD and Lookahead. We specified the eigenvalues of $A$ according to the worst-case model from Li [21] (also used by Wu et al. [43]) and set $\Sigma = A^{-1}$. We computed the expected loss (Equation 5) for learning rates in the range $(0, 1)$ for SGD and Lookahead with $\alpha \in (0, 1]$, with $k = 5$, at time $T = 1000$ (by unrolling the above dynamics). We computed the variance fixed point for each learning rate under each optimizer and use this value to compute the optimal loss. Finally, we plot the difference between the expected loss at $T$ and the final loss, as a function of the final loss. This allows us to compare the convergence performance between SGD and Lookahead optimization settings which converge to the same solution.

**Further convergence plots**    In Figure 11 we present additional plots comparing the convergence performance between SGD and Lookahead. In (a) we show the convergence of Lookahead for a single choice of $\alpha$, where our method is able to outperform SGD even for this fixed value. In (b) we show the convergence after only a few updates. Here SGD outperforms Lookahead for some smaller choices of $\alpha$. This is because SGD is able to make progress on the expectation more rapidly and reduces this part of the loss quickly — this is related to the short-horizon bias phenomenon [43]. However, even with only a few updates there are choices of $\alpha$ which are able to outperform SGD.

We also measured the optimal expected loss after some finite time horizon for both Lookahead and SGD in Figure 12. We performed a fine-grained grid search over the learning rate for SGD and both the learning rate and $\alpha$ for Lookahead (keeping $k = 5$ fixed). We evaluated 100 learning rates equally spaced on a log-scale in the range $[10^{-4}, 10^{-1}]$. For Lookahead, we additionally evaluated 50 $\alpha$ values equally spaced on a log-scale in the range $[10^{-4}, 1]$. For every time horizon, there exists settings of Lookahead that outperform SGD.

## B    Deterministic quadratic convergence analysis

Here we present additional details on the quadratic convergence analysis.

### B.1    Lookahead as a dynamical system

As in the main text, we will assume that the optimum lies at $\boldsymbol{\theta}^* = \mathbf{0}$ for simplicity, but the argument easily generalizes. Here we consider the more general case of a quadratic function $f(\mathbf{x}) = \frac{1}{2}\mathbf{x}^T A\mathbf{x}$. We use $\eta$ to denote the CM learning rate and $\beta$ for it's momentum coefficient.

Figure 11: Convergence of SGD and Lookahead on the noisy quadratic model. (a): We show the convergence of Lookahead with a single fixed choice of $\alpha = 0.4$. (b): We compare the early stage performance of Lookahead to SGD over a range of $\alpha$ values.

Figure 12: Expected loss of SGD and Lookahead with (constant-through-time) hyperparameters tuned to be optimal at a finite time horizon. At each finite horizon ($x$-axis) we perform a grid search to find the best expected loss of each optimizer. Lookahead dominates SGD over all time horizons.

First we can stack together a full set of fast weights and write the following,

$$\begin{bmatrix} \boldsymbol{\theta}_{t,0} \\ \boldsymbol{\theta}_{t-1,k} \\ \vdots \\ \boldsymbol{\theta}_{t-1,1} \end{bmatrix} = LB^{(k-1)}T \begin{bmatrix} \boldsymbol{\theta}_{t-1,0} \\ \boldsymbol{\theta}_{t-2,k} \\ \vdots \\ \boldsymbol{\theta}_{t-2,1} \end{bmatrix}$$

Here, $L$ represents the Lookahead interpolation, $B$ represents the update corresponding to classical momentum in the inner-loop and $T$ is a transition matrix which realigns the fast weight iterates.

Each of these matrices takes the following form,

$$L = \begin{bmatrix} \alpha I & 0 & \cdots & 0 & (1-\alpha)I \\ I & 0 & \cdots & \cdots & 0 \\ 0 & I & \ddots & \ddots & \vdots \\ \vdots & \ddots & \ddots & 0 & \vdots \\ 0 & \cdots & 0 & I & 0 \end{bmatrix}$$

$$
B = \begin{bmatrix}
(1+\beta)I - \eta A & -\beta I & 0 & \cdots & 0 \\
I & 0 & \cdots & \cdots & 0 \\
0 & I & \ddots & \ddots & \vdots \\
\vdots & \ddots & \ddots & 0 & \vdots \\
0 & \cdots & 0 & I & 0
\end{bmatrix}
$$

$$
T = \begin{bmatrix}
I - \eta A & \beta I & -\beta I & 0 & \cdots & 0 \\
I & 0 & \cdots & \cdots & 0 & \vdots \\
0 & I & \ddots & \cdots & \vdots & \vdots \\
\vdots & \ddots & \ddots & 0 & \vdots & \vdots \\
\vdots & \cdots & 0 & I & 0 & 0 \\
0 & \cdots & 0 & 0 & I & 0
\end{bmatrix}
$$

Each matrix consists of four blocks. The bottom left block is always an identity matrix that shifts the iterates along one index. The bottom right column is all zeros with the top-right column being non-zero only for $L$ which applies the Lookahead interpolation. The top left row is used to apply the Lookahead/CM updates in each matrix.

After computing the appropriate product of these matrices, we can use standard solvers to compute the eigenvalues which bound the convergence of the linear dynamical system (see e.g. Lessard et al. [20] for an exposition). Finally, note that because this linear dynamical systems corresponds to $k$ updates (or one slow-weight update) we must compute the $k^{th}$ root of the eigenvalues to recover the correct convergence bound.

## B.2   Optimal slow weight step size

We present the proof of Proposition 1 for the optimal slow weight step size $\alpha^*$.

*Proof.* We compute the derivative with respect to $\alpha$

$$
\nabla_\alpha L(\theta_{t,0} + \alpha(\theta_{t,k} - \theta_{t,0})) = (\theta_{t,k} - \theta_{t,0})^T A(\theta_{t,0} + \alpha(\theta_{t,k} - \theta_{t,0})) - (\theta_{t,k} - \theta_{t,0})^T b
$$

Setting the derivative to 0 and using $b = A\theta^*$:

$$
\alpha[(\theta_{t,k} - \theta_{t,0})^T A(\theta_{t,k} - \theta_{t,0})] = (\theta_{t,k} - \theta_{t,0})^T A(\theta^* - \theta_{t,0}) \tag{12}
$$

$$
\implies \alpha^* = \arg\min_\alpha L(\theta_{t,0} + \alpha(\theta_{t,k} - \theta_{t,0})) = \frac{(\theta_{t,0} - \theta^*)^T A(\theta_{t,0} - \theta_{t,k})}{(\theta_{t,0} - \theta_{t,k})^T A(\theta_{t,0} - \theta_{t,k})} \tag{13}
$$

$\square$

We approximate the optimal $\theta^* = \theta_{t,k} - \hat{A}^{-1}\hat{\nabla}L(\theta_{t,k})$, since the Fisher can be viewed as an approximation to the Hessian [25]. Stochastic gradients are computed on mini-batches used in training so as to not incur additional computational cost. Because the algorithm with fixed $\alpha$ performs so well, we only did preliminary experiments with an adaptive $\alpha$. We note that the approximation is greedy and incorporating priors on noise and curvature is an interesting direction for future work.

## C   Experimental setup

Here we present additional details on the experiments appearing in the main paper.

### C.1   CIFAR-classification

We run every experiment with three random seeds using the publicly available setup from [6]. A reviewer helpfully noted that this implementation of ResNet-18 has wider channels and more parameters than the original. For future work, it would be better to follow the original ResNet

architecture for CIFAR classification. However, we observed consistently better convergence with Lookahead across different architectures and have results with other architectures in Appendix D.3. Our plots show the mean value with error bars of one standard deviation. We use a standard training procedure that is the same as that of Zagoruyko and Komodakis [46]. That is, images are zero-padded with 4 pixels on each side and then a random $32 \times 32$ crop is extracted and mirrored horizontally $50\%$ of the time. Inputs are normalized with per-channel means and standard deviations. Lookahead is evaluated on the slow weights of its inner optimizer. To make this evaluation fair, we evaluate the training loss at the end of each epoch by iterating through the training set again, without performing any gradient updates.

We conduct hyperparameter searches over learning rates and weight decay values, making choices based on final validation performance. For SGD, we set the momentum to 0.9 and sweep over the learning rates {0.01, 0.03, 0.05, 0.1, 0.2, 0.3} and weight decay values of {0.0003, 0.001, 0.003}. The best choice was a learning rate of 0.05 and weight decay of 0.001. We found AdamW [22] to perform better than Adam and refer it to as Adam throughout our CIFAR experiment section. For Adam, we do a grid search on learning rate of {1e-4, 3e-4, 1e-3, 3e-3, 1e-2} and weight decay values of {0.1, 0.3, 1, 3}. The best choice was a learning rate of 3e-4 and weight decay of 1. For Polyak averaging, we compute the moving average of SGD use the best weight decay from SGD and sweep over the learning rates {0.05, 0.1, 0.2, 0.3, 0.5}. The best choice was a learning rate of 0.3.

For Lookahead, we set the inner optimizer SGD learning rate to 0.1 and do a grid search over $\alpha = \{0.2, 0.5, 0.8\}$ and $k = \{5, 10\}$, with $\alpha = 0.8$ and $k = 5$ performing best (though the choice is fairly robust). We report the verison of Lookahead that resets momentum in our CIFAR experiments.

## C.2 ImageNet

For the baseline algorithm, we used SGD with a heavy ball momentum of $0.9$. We swept over learning rates in the set: $\{0.01, 0.02, 0.05, 0.1, 0.2, 0.3\}$ and selected a learning rate of 0.1 because it had the highest final validation accuracy.

We directly wrapped Lookahead around the settings provided in the official PyTorch repository repository with $k = 5$ and $\alpha = 0.5$ (where SGD has a learning rate of 0.1 in the inner loop). Observing the improved convergence of our algorithm, we tested Lookahead with the aggressive learning rate decay schedule (decaying at the 30th, 48th, and 58th epochs). We run our experiments on 4 Nvidia P100 GPUs with a batch size of 256 and weight decay of 1e-4. We used the same settings in the ResNet-152 experiments.

## C.3 Language modeling

For the language modeling task we used the model and code provided by Merity et al. [27]. We used the default settings suggested in this codebase at the time of usage which we report here. The LSTM we trained had 3 layers each containing 1150 hidden units. We used word embeddings of dimension 400. Within each hidden layer we apply dropout with probability 0.3 and the input embedding layers use dropout with probability 0.65. We applied dropout to the embedding layer itself with probability 0.1. We used the weight drop method proposed in Merity et al. [27] with probability 0.5. We adopt the regularization proposed in section 4.6 in Merity et al. [27]: RNN activations have L2 regularization applied to them with a scaling of 2.0, and temporal activation regularization is applied with scaling 1.0. Finally, all weights receive a weight decay of 1.2e-6.

We trained the model using variable sequence lengths and batch sizes of 80. We apply gradient clipping of 0.25 to all optimizers. During training, if validation loss has not decreased for 15 epochs then we reduce the learning rate by half. Before applying Lookahead, we completed a grid search over the Adam and SGD optimizers to find competitive baseline models. For SGD we did not apply momentum and searched learning rates in the range {50, 30, 10, 5, 2.5, 1, 0.1}. For Adam we kept the default momentum values of $(\beta_1, \beta_2) = (0.9, 0.999)$ and searched over learning rates in the range {0.1, 0.05, 0.01, 0.005, 0.001, 0.0005, 0.0001}. We chose the best model by picking the model which achieved the best validation performance at any point during training.

After picking the best SGD/Adam hyperparameters we trained the models again using Lookahead with the best baseline optimizers for the inner-loop. We tried using $k = \{5, 10, 20\}$ inner-loop updates and $\alpha = \{0.2, 0.5, 0.8\}$ interpolation coefficients. Once again, we reported Lookahead's final performance by choosing the parameters which gave the best validation performance during training.

| Optimizer | CIFAR-10 |
|---|---|
| Maintain | $95.15 \pm .08$ |
| Interpolate | $95.16 \pm .13$ |
| Reset | $94.91 \pm .05$ |

Table 6: CIFAR Final Validation Accuracy.

Figure 13: Evaluation of maintaining, interpolating, and resetting momentum on CIFAR-10

Figure 14: Evolution of test accuracy on CIFAR-10 and ImageNet.

For this task, $\alpha = 0.5$ or $\alpha = 0.8$ and $k = 5$ or $k = 10$ worked best. As in our other experiments, we found that Lookahead was largely robust to different choices of $k$ and $\alpha$. We expect that we could achieve even better results with Lookahead if we jointly optimized the hyperparameters of Lookahead and the underlying optimizer.

### C.4 Neural machine translation

For this task, we trained on a single TPU core that has 8 workers each with a minibatch size of 2048. We use the default hyperparameters for Adam [42] and AdaFactor [40] in the experiments. For Lookahead, we did a minor grid search over the learning rate $\{0.02, 0.04, 0.06\}$ and $k = \{5, 10\}$ while setting $\alpha = 0.5$. We found learning rate 0.04 and $k = 10$ worked best. After we train those models for 250k steps, they can all reach around 27 BLEU on Newstest2014 respectively.

## D Additional Experiments

### D.1 Inner Optimizer State

Throughout our paper, we maintain the state of our inner optimizer for simplicity. For SGD with heavy-ball momentum, this corresponds to preserving the momentum. Here, we present a sensitivity study by comparing the convergence of Lookahead when maintaining the momentum, interpolating the momentum, and resetting the momentum. All three improve convergence versus SGD.

### D.2 Validation Accuracy

We present curves corresponding to the evolution of validation accuracy during training on CIFAR and ImageNet datasets. Though not the focus of our work, we find that faster convergence in training loss does in fact correspond to better validation performance on these datasets.

### D.3 Comparison to Stochastic Weight Averaging

In this subsection, we elaborate on differences between Stochastic Weight Averaging (SWA) [14] and Lookahead, showing that they serve different purposes but can be complementary. First, SWA and the general family of methods that perform tail averaging [36, 34] requires a choice of when to begin averaging. A choice that is either too early or too late can be detrimental to performance. This is illustrated in Figure 15, where we plot comparisons of the test accuracy of three runs of Lookahead and SWA, using SGD in the inner loop of both algorithms. We use the suggested schedule from the SWA paper, which has higher learning rates than is typical at the end of training. SWA achieves better

Figure 16: Test Accuracy on CIFAR-100 with SWA and Lookahead (Wide ResNet-28-10). Following Izmailov et al, SWA is started at epoch 161. We plot the accuracy throughout training (left) and the accuracy of the SWA network (right).

performance when initialized from epoch 10 compared to epoch 1, due to poor performance from the earlier models. In contrast, Lookahead is used from initialization and does not have this tail averaging start decision. While SWA performs better during the intermediate stages of training (since it is loosely approximating an ensemble by averaging the weights of multiple models), Lookahead with its variance reduction properties achieves better final performance, even with the modified learning rate schedule. Lookahead also computes an exponential moving average of its fast weights rather than the aritthmetic average of SWA, which increases the emphasis on recent proposals of weights [25].

We do believe that Lookahead is complementary to SWA and traditional techniques for ensembling models. To this end, we perform three runs with Wide ResNet-28-10 [46] on CIFAR-100 with SWA and compare two choices of the inner loop algorithm of SWA. The inner loop algorithms are SGD (as in [14]) and Lookahead wrapped around SGD. The runs with Lookahead achieve higher test accuracy throughout training and in the weight averaged network.

Figure 15: Test Accuracy on CIFAR-100 with SWA and Lookahead (PreResNet-110). We follow exactly the hyperparameter settings in their repository and also run Lookahead with $\alpha = 0.8$ and $k = 10$. Note that the learning rate schedule uses a learning rate that is higher than is typical at the end of training.