[Reviews · NeurIPS 2019]

Reviewer 1



UPDATE: After reading the rebuttal and discussing with the other reviewers, I'd like to increase my score to 5. Particularly I think that the theoretical part is weak, and it needs significant improvement. %%%%%%%%%%%%%%%%%%%%%%%%%%%%%%%%%%%%%%%%%%%%%%%%%%%% This paper presents a Lookahead algorithm for optimizing deep neural nets. The algorithm consists of an inner loop, which updates fast weights using conventional optimizers like SGD, and an outer loop, which updates slow weights by linear interpolation. Also some theoretical analysis on optimal learning rate and convergence is provided as well. The paper is written clearly, easy to follow. My main concerns about the paper are as follows: 1. Novelty: The proposed idea has been explored in Zhang et al. 2018 (https://www.merl.com/publications/docs/TR2018-206.pdf) where they proposed an algorithm called Time-Delay momentum, equivalent to fast/slow weights, though I think there may be some errors in their theoretical analysis. Similar performance such as training and testing behavior on image classification was reported as well. 2. I do not see any usage about Sec. 2.1, as stated in L91 that in practice the authors are still using fixed learning rate. So what is the benefit of providing such a result? 3. In Yanet al., Unified analysis of stochastic momentum methods for deep learning, IJCAI 2018, the convergence of momentum in expectation was proved for (convex) deterministic loss functions. Sec. 3.2 verifies such a result. 4. The empirical results are weak, and I do not see the benefit of using the proposed method over traditional ones like SGD and Adam.

Reviewer 2



Update: I have read the author's response and have kept my score. Please note that in DeVries and Taylor'17, 'ResNet-18' is not truly the ResNet-18 model (it consists of 4 stages and has more than an order of magnitude more parameters than the original ResNet-18 due to wider channels). This should be made clear in the paper in order not to cause more confusion in the community. Originality: Medium/High The proposed algorithm is considerably different than recently proposed methods for deep learning, which gravitate towards adaptive gradient methods. It has some similarities to variance reduction algorithms with inner and outer loops, however Lookahead has a very simple outer loop structure and and is easy to implement. I consider it a significantly original work. Quality: Medium The experimental evaluation is very extensive, including many challenging tasks, and improvement on language modelling is significant. The analysis, although helps motivate the proposed algorithm, is restricted to a very simplistic setting -- any analysis for the smooth non-convex case would be very helpful since the method is aimed at training neural networks, even if it provided no advantage over SGD. The CIFAR experiments, however, look a bit off. To achieve 4.7-4.8% error on CIFAR-10 a very deep ResNet is typically requried (a ResNet-1001 achieves similar performance, as reported in [1]), and I find it unlikely that the authors achieved such performance with a ResNet-18. I would guess that a Wide ResNet was used instead of a standard ResNet model, with a considerably widening factor, but I expect this to be addressed in the rebuttal. Clarity: High The paper is clear and well written. Significance: Medium/High The paper manages to successfully convince that variance reduction-type methods with simple inner/outer loops can provide significant advantages to deep learning applications, and might increase interest in such kind of research. The idea is refreshing in face of how much focus is aimed on adaptive gradient methods and Adam-type variants currently, and the paper shows through extensive experiments that Lookahead can provide performance boosts when compared to other methods -- especially in tasks where adaptive methods outperform SGD (in the paper, language modelling and machine translation).

Reviewer 3



*****************After Author Response*************** Dear authors, thank you very much for your response. After much consideration, I find that the response of the authors does address my initial concerns. Though after much deliberation, I believe the numerics are convincing and the contribution is clear. Thus I have increased the grade to 6. I kindly request that the authors make a clear comparison with SWA, both on the high level and experimentally. In particular, I was not satisfied with the authors response because: 1) (major) I pointed out in my review that LA and SWA are similar augmentations by writing SWA in the same format as LA. Though I agree they are not the same, they are at least competing methods. The authors responded stating that SWA and LA "serve different purposes and are complementary". The justification for this is that LA uses and exponentially moving average and SWA was is typically initiated towards the end. This for me is a justification of why they are not the same method (which I already agreed), but it does clarify why they are not directly competing augmentations (note that SWA could be applied throughout). Though I am pleased the authors offered more experiments applying LA in conjunction with SWA, I think if makes more sense to directly compare the performance of SGD + LA and SGD + SWA to best verify what is the contribution of each method. 2) (minor) In my review I pointed out that the theory in Section 3.1 is very restrictive since optimizing a diagonal quadratic is equivalent to minimizing several decoupled 1-dimensional second order polynomials. The authors responded stating that it "is equivalent to general stochastic convex quadratics" pointing to Sutskever et al. 2013, Proposition 6.1 where the authors show that the analysis of *full* gradient descent with/without momentum depends only on the eigenvalues of the quadratic. But this is not true for stochastic gradient methods. Because for quadratic SGD can be cast as a method that samples rows or columns of the quadratic matrix. But what is a row or column depends on the coordinate basis, and as a result, the analysis of SGD does depend on eigenvalues and eigenvectors. Thus the authors response is in accurate. Though I do concede that, it sheds light on the convergence of full batch gradient methods combined with the lookahead scheme. ********************************************************** *Issues* Similarity to Stochastic weight averaging: The proposed lookahead algorithm is similar to the Stochastic Weight Averaging (SWA) [1], so much so that SWA could be considered as a variant of Lookahead, and vice-versa. To see this, I have re-written the SWA Algorithm in the same notation used in this paper: SWA Algorithm: 1. phi_{0} = O_{1,0} = initial parameter 2. for t = 1,2, … do 3. for t = 1,2, …, k do 4. O_{t,i} = O_{t,i-1} + A(L, O_{t,i-1}, d) 5. end for 6. alpha_t = 1/(t+1) 7. phi_t = phi_{t-1} + alpha_t (O_{t,k} - phi_{t-1}) 8. end for Note that I re-arranged the notation by using two loops (as done here) as opposed to using the mod function (as used in [1]). The difference between the SWA algorithm and the Lookahead is that the inner iterations are not reset using the outer iterations (there is no O_{t,0} = phi_{t-1} step as there is in Lookahead) and the parameter alpha depends on the outer iteration counter (see line 6 of SWA Algorithm). Excluding these two difference, the methods are essentially the same. Furthermore, both rely on the same intuition and both paper make use of the same projected level set plots (Figure 1 in this paper and Figure 1 in the SWA paper [1]). Finally both have the same computation overhead, are easy to implement, and are directly competing augmentations (it doesn’t make sense to apply both independently). This is why I feel the Lookahead procedure should have been compared to SWA and these similarities drawn out. Though I agree the other augmentations mentioned in Section 4 are not directly competing since they can be applied in conjunction with Lookahead (variance reduction, acceleration, Anderson averaging ...etc ). Indexing mistake eq (1) and (2): There is a small indexing mistake here, instead of Equation (1) according to Algorithm 1 we should have \phi_{t+1} = \phi_t + \alpha (\theta_{t+1,k} - \phi_t) Thus unrolling this recurrence gives \phi_{t+1} = (1-\alpha)^{t+1} \phi_0 + \alpha\sum_{i=0}^{t+1}(1-\alpha)^i\theta_{t+1-i,k} Line 72: … operations that amortized … Proposition 1: This proposition has a few issues. The formula for \hat{\alpha}^* has not been explained anywhere (nor in the appendix). In particular, is \nabla L(O_{t,k}) the full gradient or a stochastic gradient (there is no mini-batch notation). What is the motivation behind using O_{t,k} - \hat{A}^{-1} \nabla L(O_{t,k}) in place of O^*? Also you should explicitly define that \hat{A} is this diagonal of an approximate empirical Fisher matrix mentioned on line 84. Finally, it seems this entire Section 2.1 could be removed since on lines 91 – 94 you explain that this approximation does not offer any significant gain to using a fixed alpha. Theory in Section 3.1. The model problem is very restrictive. Indeed, having a diagonal quadratic is equivalent to minimizing several decoupled 1-dimensional second order polynomials. Thus Proposition 2 essentially shows how the lookahead affects the asymptotic fixed point of the variance on 1-dimensional 2nd order polynomials. It is often misleading to draw conclusions of how a method behaves by analyzing it in 1-dimension (e.g. efficient methods for solving 1-dimensional optimization problems are very different as compared to the efficient methods for solving n-dimensional problems). If instead the authors had analyzed a convex quadratic (not necessarily diagonal), this would have been much more informative. Line 129: Why measure the rate of contraction r:= 1.0 - ||O_t ||/ ||O_{t-1} || in the plots? If r_t → 0 this does not show that the methods converges. Why not report the decrease in the expected quadratic loss function instead? Line 153: The description of the SWA “… (SWA), which averages the weights of different neural network obtained during training” seems to be incorrect. Indeed, as I detailed before, SWA is very much the same type of augmentation as Lookahead. There is only one neural network. Line 177: … their method requires the order of …. Section B.1: The notation for the matrix A is overloaded, making it harder to follow this section. Note that A is at the same time the matrix in the quadratic x^T A x and it is the matrix that represents the lookahead interpolation. Please introduce different symbols for these two matrices. Line 439: “we set … learning rate of {0.1, 0.2}”. This seems to be a mistake. What was the value chosen for the learning rate? [1] Pavel Izmailov, Dmitrii Podoprikhin, Timur Garipov, Dmitry P. Vetrov, Andrew Gordon Wilson: Averaging Weights Leads to Wider Optima and Better Generalization. UAI 2018: 876-885

[Author Response · NeurIPS 2019]

Thank you for all the helpful comments. Several related works were raised by the reviewers which we discuss here.

**Time-Delay Momentum:** We note that the authors have marked their ArXiv submission as containing errors. While their algorithm also utilizes an inner-outer loop structure, it differs significantly from ours in motivation and implementation. Each of their inner loops uses SGD to solve the *distance-regularized* objectives. We instead allow arbitrary optimization algorithms in the inner-loop and solve the *original objective*. Our main contribution is introducing interpolation to update both the slow and fast weights. This is a critical difference which leads to an exponential moving average (EMA) with advantages such as variance reduction [Martens 2014, Polyak and Juditsky, 1992].[1] **SWA:** While both SWA and Lookahead average network weights, we believe they serve different purposes and are complementary. First, we use the EMA of slow weights to adjust the training parameters during optimization. Compared to using the EMA at inference, this gives much faster convergence via variance reduction along the trajectory (see comparison to Polyak averaging in Figure 5). Second, SWA is applied near convergence, whereas Lookahead is applied throughout training. This removes the challenge of deciding when to start iterate averaging. We include an evaluation with SWA in Figure 1. We use 3 random seeds with publicly available code and hyperparameters[2] from Izmailov et al. to show that Lookahead and SWA are complementary. Lookahead wrapped around SGD dominates SGD in performance during training and improves the weight averaged network.[3] We observe similar behavior on ResNet-110 and VGG-16.

**Reviewer 3   Adaptive Learning Rate Result** We believe this is an interesting result but agree that it is not critical and do not mind excluding it.
**SUM Framework** Yan et al. prove results for non-convex stochastic losses. Furthermore, Lookahead does not fit into their SUM framework. We do not believe that this work is relevant to Section 3.2 or otherwise strongly related to our own.
**Strength of empirical results** We respectfully disagree. It is difficult to improve on the convergence and final performance of the carefully tuned baseline methods, which is why they have remained the primary choices for so long. Despite this, we show significant wins in terms of convergence and final performance over a range of different tasks. For language modelling we provide 1.5 perplexity improvement with much faster convergence (Table 3). On image classification tasks we are consistently able to achieve better final performance in fewer update steps. We also show that Lookahead is very robust to different inner-optimizer settings (Figure 8). A method that reduces hyperparameter tuning, improves convergence, and strictly[4] improves final performance is useful for the community, especially to resource-constrained groups.

Figure 1: Test Accuracy on CIFAR-100 with SWA and Lookahead (Wide ResNet-28-10). Following Izmailov et al., SWA is started at epoch 161.

**Reviewer 5**   We used the ResNet-18 architecture, which was designed for ImageNet but has been used for CIFAR in work such as DeVries and Taylor [2017], with which our numbers agree.

**Reviewer 6**   We have corrected typos in Proposition 1 and Algorithm 1 and fixed other minor mistakes.
**On NQM model**: This model has been studied in prior work [Martens 2014, Wu et al., 2018, Lucas et al., 2018] and **is** equivalent to general stochastic convex quadratics under the studied schemes (see Sutskever et al. 2013, Proposition 6.1). This system is **not** equivalent to decoupled 1-D problems as the learning rate is shared over all dimensions. Furthermore, [Zhang et al., 2019] recently showed that insights from this NQM capture many essential features of neural net training.
**LA inner learning rate for CIFAR:** We used $0.1$ in the reported results except Figure 8.
**Rate of contraction**: We will replace this with the decrease in the loss though the same conclusions apply. The linear dynamical system admits a unique fixed point at the minima (origin) for both lookahead and CM — thus the rate of contraction is a valid measure of convergence (see Lessard et al., 2014).
**SWA multiple networks**: We mean that different neural network weights are added to the moving average during training and this is interpreted (loosely) as ensembling. We will fix the wording here, thank you.
**Additional references:** Terrance DeVries and Graham W Taylor. Improved regularization of convolutional neural networks with cutout. *arXiv preprint arXiv:1708.04552*, 2017.
Guodong Zhang, Lala Li, Zachary Nado, James Martens, Sushant Sachdeva, George E Dahl, Christopher J Shallue, and Roger Grosse. Which algorithmic choices matter at which batch sizes? insights from a noisy quadratic model. *arXiv preprint arXiv:1907.04164*, 2019.

## Footnotes

[1]Citations not in references below are in the original manuscript.

[2]200 epochs of training with SWA initalized from epoch 161, initial LR 0.1, SWA LR 0.05 and weight decay of $5 \times 10^{-4}$

[3]Note that we do not tune the learning rate schedule for Lookahead and follow the schedule proposed by Izmailov et al. The learning rate is higher than is typical at the end of training, which explains the large gap to the non-SWA performance.

[4]We did not find tasks for which Lookahead performed worse than its inner optimizer, unlike adaptive vs non-adaptive gradient-based optimizers.


[Meta-Review · NeurIPS 2019]

The theoretical analysis received much criticism during the discussions. Parts of the discussion have been updated in the reviews. Overall, the theoretical contributions are weak and distracting from the main paper. We suggest either strengthening the section or moving it to a less prominent position. In particular, we have ignored the theoretical contributions in the evaluation. A comment from the senior area chair on the theoretical analysis is quoted below. "The "analysis" is very disappointing, limited and I don't think particularly insightful. Its only for restricted types of quadratic objectives, which is very limited. And for stochastic objectives the claim is that for a fixed step size the radius of convergence is smaller (recall that with a fixed step size SGD will not converge to the optimum, but only to a radius around the optimum). But this is not really what we care about---we care about how quickly it gets there. You can easily change this radius by changing the step size." While the majority of the reviewers have found the experimental contributions convincing, there are some lingering concerns. Major concerns: 1. Tuning step size for baselines: The authors provide the tuning parameters used in the experiments in the appendix, but do not state what the final value that was chosen in the results reported in Section 5 are. In particular, the extensiveness of grid-search for SGD step size is crucial here. Please include experiments addressing the following: (a) Grid search should include values that are "signficanlty larger" as well as "significantly smaller" than the optimal parameter chosen for final numbers. (b) At least for smaller datasets, a finer grid search should be used. 2. Comparison to SWA: Even though SWA in original paper is proposed as fine tuning method (on last few epochs), Reviewer #5 has clearly laid out (in the initial review) the similarities to the current method and how it can be used throughout training (rather than just fine tuning on last few epochs). The authors seem to have missed this point in the response and continue to use SWA as a fine-tuning tool. We recommend revisiting the algorithm and adding empirical comparison to SGD+SWA where SWA is used *throughout* training - this should look like skipped-polyak-averaging where every 'k'th iterates are averaged. See reviewer #5's updated comment. 2. Test accuracies across epochs: Reviewer #3 rightly points out the Fig 5-7 show faster convergence only in training loss. Although Tables 2-3 show improvements in test scores, it would be quite useful to plot test accuracies across epochs (analogues of Fig 5-7 with test accuracies) Minor concerns: 1. Figure 3: although this figure is not the main contribution, it is not parse-able. Please provide appropriate details on what is "range of values of alpha" and how they correspond in the plot? what is dark blue vs light blue lines in the plot? 2. Figure 2: What precisely is plotted on y-axis? If it is average training cross-entropy loss, it is surprising that the loss is so low at initialization (or even at 1 epoch)? Minor: Fig 5 title is wrong. Please fix other typos in the paper carefully.